# Aerotaxis in the closest relatives of animals

**Julius B Kirkegaard, Ambre Bouillant, Alan O Marron, Kyriacos C Leptos, Raymond E Goldstein\***

Department of Applied Mathematics and Theoretical Physics, Centre for Mathematical Sciences, University of Cambridge, Cambridge, United Kingdom

**Abstract** As the closest unicellular relatives of animals, choanoflagellates serve as useful model organisms for understanding the evolution of animal multicellularity. An important factor in animal evolution was the increasing ocean oxygen levels in the Precambrian, which are thought to have influenced the emergence of complex multicellular life. As a first step in addressing these conditions, we study here the response of the colony-forming choanoflagellate *Salpingoeca rosetta* to oxygen gradients. Using a microfluidic device that allows spatio-temporal variations in oxygen concentrations, we report the discovery that *S. rosetta* displays positive aerotaxis. Analysis of the spatial population distributions provides evidence for logarithmic sensing of oxygen, which enhances sensing in low oxygen neighborhoods. Analysis of search strategy models on the experimental colony trajectories finds that choanoflagellate aerotaxis is consistent with stochastic navigation, the statistics of which are captured using an effective continuous version based on classical run-and-tumble chemotaxis.

**\*For correspondence:** R.E.
Goldstein@damtp.cam.ac.uk

**Competing interests:** The
authors declare that no
competing interests exist.

**Reviewing editor:** Richard M
Berry, University of Oxford,
United Kingdom

## Introduction

*Taxis*, the physical migration towards preferred or away from undesired conditions, is a feature shared by virtually all motile organisms. Taxis comes in many forms, and in common is an underlying field of attractant (or repellent) and an ability to react and navigate along gradients of this field. Bacteria do *chemotaxis* towards nutrients (*Adler, 1969*; *Berg, 1993*) and away from toxins (*Tso and Adler, 1974*). Algae do *phototaxis* towards light (*Yoshimura and Kamiya, 2001*; *Drescher et al., 2010*) and *gyrotaxis* along gravitational potentials (*Kessler, 1985*). Chemotaxis provides a mechanism for the recognition and attraction of gametes (*Vogel et al., 1982*) and for complex behavioural patterns such as in the slime mould *Dictyostelium discoideum*, where cAMP-driven chemotaxis is a critical part of the formation of the multicellular stage of the life cycle (*Bonner, 1947*). Aerotaxis, defined as oxygen-dependent migration, is well-characterized in bacteria (*Taylor et al., 1999*), but is poorly studied in more complex organisms. This is despite the essentiality of oxygen for all aerobic life, and the important role that Precambrian oxygen levels played in the emergence and evolution of multicellular animal life (*Nursall, 1959*).

One group of aquatic heterotrophic protists, the choanoflagellates, are of particular interest for the study of how multicellularity evolved. Choanoflagellates are a class of unicellular microorganisms that are the closest relatives of the animals (*Lang et al., 2002*). This relationship was first proposed by James-Clark in 1866 (*James-Clark, 1866*), on the basis of the resemblance between choanoflagellates and the choanocytes of sponges. The sister relationship between choanoflagellates and animals was further confirmed in the genomic era by molecular evidence (*King et al., 2008*). All choanoflagellates share the same basic unicell structure: a prolate cell body with a single beating flagellum that is surrounded by a collar of microvilli. The beating of the flagellum creates a current in the surrounding fluid that guides suspended prey such as bacteria through the collar (*Pettitt and*

**eLife digest** Most animals are made up of millions of cells, yet all animals evolved from ancestors that spent their whole lives as single cells. Today the closest single-celled relatives of animals are a group of aquatic organisms called choanoflagellates. Certain species of choanoflagellates can also form swimming colonies. This kind of multicellularity might resemble that seen in the earliest of animals. As such, studies into modern-day choanoflagellates can give insights into how the first animals to evolve might have behaved.

Many organisms can find their way towards favorable areas using different strategies. For instance, bacteria can bias their tumbling to gradually swim towards food, and algae can turn and move directly towards light. While choanoflagellates require oxygen, it was not known if they could also actively navigate towards it, or any other resource.

Now, Kirkegaard et al. find that the choanoflagellate *Salpingoeca rosetta* can indeed navigate towards oxygen – an ability called aerotaxis. This was true for both individual cells and for colonies made up of many cells. This discovery suggests that the transition from living as a single cell to living as a simple multicellular organism could still have allowed the earliest animals to seek out and move towards resource-rich areas.

Aerotaxis requires cells to both sense oxygen and react appropriately to changes in its concentration. Kirkegaard et al. watched choanoflagellate colonies swimming under controlled conditions and varied the oxygen concentration in the water over time. These experiments revealed that the colonies navigate based on the logarithm of the oxygen concentration, so that at low oxygen levels the cells were even more sensitive to small changes in oxygen concentration. This type of 'logarithmic sensing' is similar to how our ears sense sounds and our eyes sense light. Kirkegaard et al. went on to conclude that the colonies were not actively steering in the correct direction directly. Instead, the colonies appeared to choose directions at random and later decide whether such a turn was correct.

It remains unclear whether the common ancestor of animals and choanoflagellates could also perform aerotaxis, and if so what mechanisms this involved. Further studies to compare aerotaxis and aerotaxis-related genes in simple animals and other single-celled relatives of animals would be needed to illuminate this. Future studies could also explore the maximum and minimum oxygen concentrations that choanoflagellates can detect, and how well they navigate at these upper and lower limits.

Orme, 2002) where they can be caught and ultimately phagocytosed. The flagellar current also has the effect of causing the choanoflagellate cell to swim.

The choanoflagellate *Salpingoeca rosetta* can form colonies through incomplete cytokinesis (*Fairclough et al., 2010*). In the presence of certain bacteria (*Dayel et al., 2011*; *Levin et al., 2014*), these colonies have an eponymous rosette-like shape as shown in *Figure 1*. The colony morphology is variable, and the constituent flagella beat independently of one another (*Kirkegaard et al., 2016*). The random and independent flagellar motion argues against there being any coordination between cells in a colony, and as yet no evidence of any form of taxis for choanoflagellate colonies has been reported.

The geometry, flagella independence and lack of taxis observed in *S. rosetta* colonies contrast with other lineages, such as the Volvocales, a group of green algae (*Goldstein, 2015*). Phototaxis is clearly observable in both unicellular (*Chlamydomonas*) (*Yoshimura and Kamiya, 2001*) and colonial (*Volvox*) (*Drescher et al., 2010*) species, in order to maintain optimum light levels for photosynthesis. Volvocalean phototaxis is *deterministic*, requiring precise tuning between the internal biochemical timescales and the rotation period of the organism as a whole. Although *S. rosetta* colonies also rotate around an internal axis, due to the variable colony morphology and the independent beating of the individual flagella, this rotation rate will itself be random (*Kirkegaard et al., 2016*), rendering a strategy similar to that of the Volvocales unlikely in *S. rosetta*.

An alternative strategy is *stochastic* taxis, sometimes referred to as kinesis. The classic example of stochastic taxis is the run-and-tumble chemotaxis of certain peritrichous bacteria (*Berg, 1993*). By

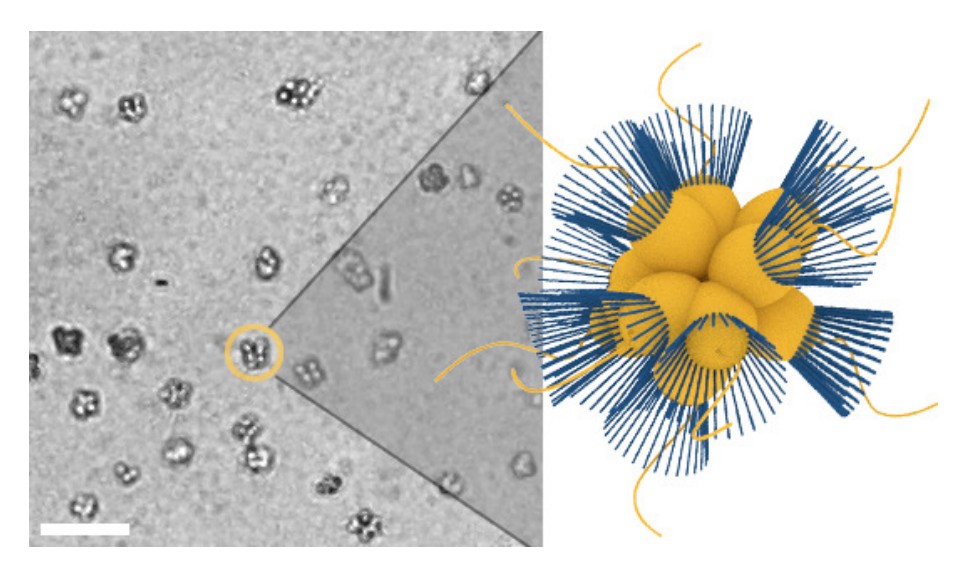

**Figure 1.** Micrograph of *S. rosetta* colonies (left) with schematic illustration (right, collars in blue). Scale bar: 50 μm. Cell body diameters are ~5 μm.

spinning their left-handed helical flagella in different directions, such bacteria can alternate between swimming in straight lines (running) and randomly reorienting themselves (tumbling). Through biasing tumbles to be less frequent when going up the gradient, the bacteria exhibit biased motion towards a chemoattractant without directly steering towards it (*Berg, 1993*).

Here, we study *S. rosetta* and show that it exhibits aerotaxis, *i.e.* navigation along gradients of oxygen. We further examine and statistically analyse aerotaxis of *S. rosetta* colonies under spatio-temporal variations of oxygen at the level of total colony populations and at the level of the trajectories of individual colonies. From these experiments we establish two key features of the aerotactic response of choanoflagellates: they employ a stochastic reorientation search strategy and the sensing of oxygen concentration gradients is logarithmic. Finally, we render these results quantitative through the use of mathematical analysis of a modified Keller-Segel model (*Keller and Segel, 1971*).

## Results

### Experimental set-up

The study of aerotaxis in bacteria has led to numerous methods for creating spatial oxygen gradients (*Shioi et al., 1987*; *Wong et al., 1995*; *Zhulin et al., 1996*; *Taylor et al., 1999*), one of which is the exploitation of soft lithography techniques (*Adler et al., 2012*; *Rusconi et al., 2014*). Since PDMS, the most commonly used material for microfluidic chambers, is permeable to gases, gas channels can be introduced in the devices to allow gaseous species to diffuse into the fluid. For example, an oxygen gradient can be created using a source channel flowing with normal air and a sink channel flowing with pure nitrogen.

Our device, shown schematically in *Figure 2*, is a modified version of that used by Adler, *et al.* (*Adler et al., 2012*). Viewed from above, the sample channel (yellow) consists of a wide observation chamber with thin inlet and outlet channels. The outlet leads to a serpentine channel that hinders bulk fluid flows. On each side of the sample channel are gas channels, the inlets of which are connected to a valve system allowing for the flow of air (20% oxygen) and nitrogen. The flow of air and nitrogen can be conducted in any combination and configuration, *e.g.* oxygen in one channel and nitrogen in the other, and can be easily swapped over. The PDMS chamber is plasma etched to a glass slide, and an extra glass slide is etched on top of the device, preventing air from diffusing in from the surrounding environment.

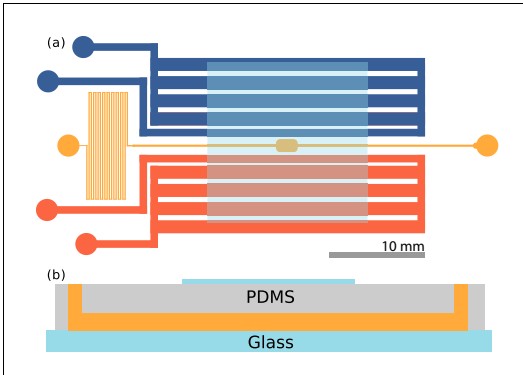

**Figure 2.** Microfluidic device. (**a**) Top view of the device. The sample channel (yellow) is loaded with culture and observed in the middle chamber. The side channels (red, blue) are gas channels in which oxygen and nitrogen may be flown. Scale bar: 10 mm. (**b**) Side view of the device. PDMS is plasma etched to a glass slide, and a cover slip is plasma etched on top, centered on the imaging chamber, also shown in (**a**). Thickness of the channels are $\approx 115 \, \mu m$.

Experiments were carried out immediately after plasma etching as the permeability of gases slowly decreases thereafter. Cultures of *S. rosetta* were introduced at the inlet of the device, and both the inlet and outlet were then closed to prevent evaporative flows. A gradient of oxygen was set up by having air flowing in one of the side channels and nitrogen in the other.

## Aerotaxis in choanoflagellates

Our main experimental result, shown in *Figure 3a,b*, is the observation that *S. rosetta* colonies accumulate at the oxygen-rich side and away from the oxygen-poor side, *i.e.* that they are aerotactic. We also found aerotaxis in the unicellular fast swimmer form (*Dayel et al., 2011*) of *S. rosetta*, showing that this is not an exclusive phenomenon to colonies.

With the present microfluidic device we can explore more details of choanoflagellate aerotaxis by dynamically changing the oxygen boundary conditions, for instance by flipping the gradient direction or by removing all oxygen influx after a uniform distribution has been reached. *Figure 3c* shows the result of such a dynamic experiment over the course of ∼3.5 hr. The density is normalized for each frame and the noise present is partly due to colonies missing in the tracking in some frames. Many repetitions of the experiment show that the behaviour in *Figure 3c* is highly repeatable and robust to changes in the details of the cycling protocol (See *Figure 3—figure supplement 1*). For consistency the figures in the main text are based on this specific protocol.

Whenever one gas channel contains oxygen and the other nitrogen, the colonies swim towards the oxygen-rich side as further shown in *Video 1* . In the time after a gas channel swap, the slope of the maximum density reveals the ensemble drift velocity $v_{\mathrm{drift}}$. When there is oxygen in both gas channels, we observe that the density reaches an approximately uniform distribution within the time frame of the experiment. For periods in which nitrogen flows in both channels, this is not the case. Under these experimental conditions, the colonies accumulate in the middle of the chamber, where there is still some residual oxygen, as further shown in *Video 2*. The fact that in this nitrogen-only configuration the colonies accumulate mid-chamber shows that accumulation does not depend on the presence of a nearby surface. With only nitrogen flowing, eventually there will be no oxygen gradient. Nonetheless, we observed the colonies to stay in the middle of chamber even after 90 min (see *Figure 3—figure supplement 1*). At that time, the highest oxygen levels are estimated by the diffusion equation to be less than ∼0.2%.

This contrasting behaviour between the oxygen-only and nitrogen-only configurations suggests an asymmetry or non-linearity in the aerotactic response. If the response to oxygen concentration had been linear, the observation of the density band in the nitrogen-only section would imply similar density bands at the chamber edges in the oxygen-only section, which is not observed. Instead one might hypthesize that the colonies navigate along *relative* ($\nabla c / c$) instead of *absolute* ($\nabla c$) gradients, *i.e.* reacting to gradients that are comparable in magnitude to the background concentration. This is also known as logarithmic sensing, and we will confirm in the modelling section that this hypothesis can quantitatively explain the experiments.

## Navigation strategy

Strategies of taxis can be categorized into two main classes: *deterministic* and *stochastic*. In both strategies the swimming organism measures the attractant gradient (for small organisms by some temporal filter [*Block et al., 1982*; *Celani and Vergassola, 2010*]). A deterministic strategy, then, is one in which

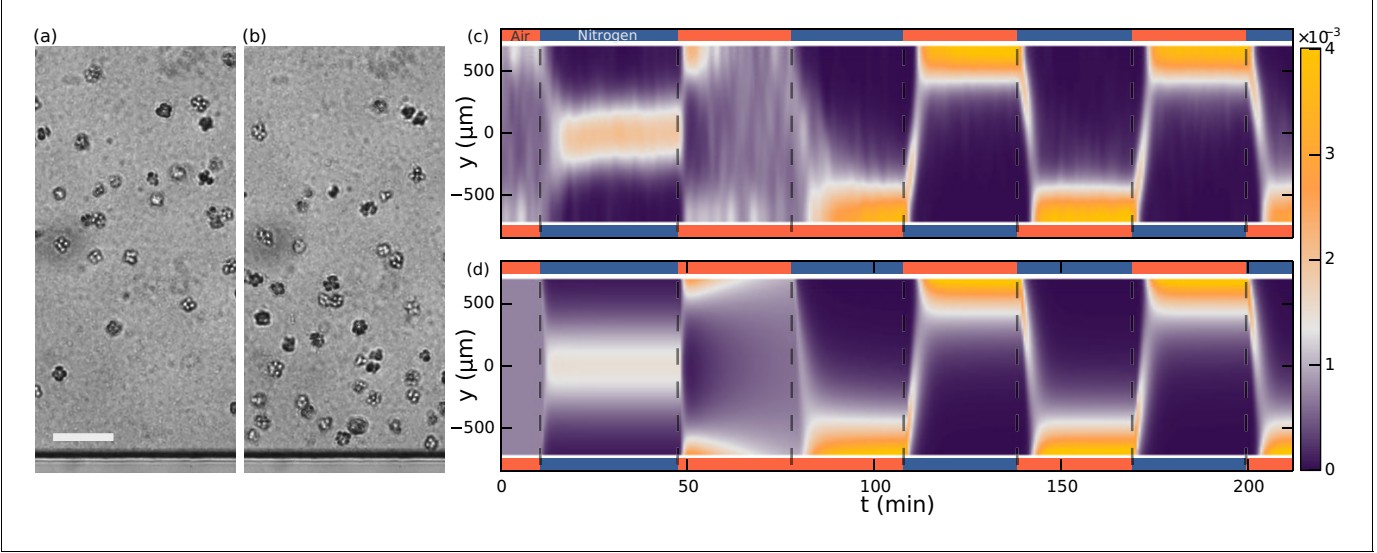

**Figure 3.** Aerotaxis of *S. rosetta* colonies. (a–b) Micrographs near an oxygen-rich wall at twice the resolution of that used in the density experiments. Scale bar: 50 μm (a) Colonies approach a wall where the oxygen-concentration is high. (b) Colonies staying near this wall. (c) Density evolution of *S. rosetta* during experiment. At each time step the distribution is normalized to a probability distribution [colorbar units in μm$^{-1}$]. Colors on the side indicate what gas is flowing in that side channel, red for oxygen and blue for nitrogen. $N_{colonies}$~150, concentration ~5 · 10$^6$ mL$^{-1}$. (d) Keller-Segel model with log-concentration input given by *Equation 4*, $D$ = 865 μm$^2$/s, $\alpha$ = 1850 μm, $v_{drift}$ = 5.2 μm/s.

The following figure supplement is available for figure 3:

**Figure supplement 1.** Seperate aerotaxis experiment.

the organism directly steers towards the attractant, such as seen in sperm cells that modulate their flagellar beat to adjust directly the curvature and torsion of its swimming path in the gradient direction (*Friedrich and Jülicher, 2007*; *Jikeli et al., 2015*). Contrasting is a stochastic strategy such as bacterial run-and-tumble locomotion (*Berg, 1993*), where modulation of the frequency of random reorientations biases the motion in the gradient direction without directly steering towards it.

One simple method of taxis results from an organism swimming faster when it is moving up the gradient, creating an overall bias towards the attractant. With the detailed colony-tracking in the present study it is possible to test whether this mechanism is in operation with *S. rosetta*. *Figure 4—figure supplement 1* shows examples of tracks during periods of uniform swimming (*t* = 70 min) and after a gas channel swap (*t* = 142 min). *Figure 4* shows the evolution of the mean colony swimming speed *v* (green) as well as the component velocities $v_x$ (yellow) and $v_y$ (purple), averaged over ~150 colonies in each frame. For most times, the component velocities average to zero, but after a gas channel swap the *y*-component peaks. The ensemble average swimming speed in these sections, however, does not show an increase, suggesting that a velocity modulation is not the method of taxis. To quantify this further, the inset of *Figure 4* shows the swimming speed in these sections plotted against the alignment to the gradient $\hat{c}\,v_y/v$ where $\hat{c} = \pm 1$ signifies the direction of the gradient. The plot shows a very small (~3%) change in swimming speed going up the gradient. Velocity-biased taxis can be described by $\mathbf{v}(t) = v(\hat{p})\,\hat{p}$, where *e.g.* $v(\hat{p}) = v\,(1 + \gamma\hat{p} \cdot \nabla c/|\nabla c|)$, $\gamma$ being the velocity-modulation taxis parameter. $\hat{p}$, the direction of swimming, is unbiased by the attractant field *c* and evolves by rotational diffusion. To obtain a drift velocity ~1/3 of the swimming velocity, as we find for *S. rosetta* in the following section, the velocity modulation would have to be $\gamma$= 2/3 for a two-dimensional swimmer and $\gamma$ = 1 in three dimensions, much larger than the ~3% observed. We conclude that the primary mechanism of aerotaxis in *S. rosetta* is therefore not a modulation of swimming speed.

*S. rosetta* colonies swim along noisy helical paths, and each colony displays distinct helix parameters (*Kirkegaard et al., 2016*). To perform any kind of statistical angle analysis, we consider ensemble average quantities: the speed *v* and rotational diffusion $d_r$, and average helix rotations out.

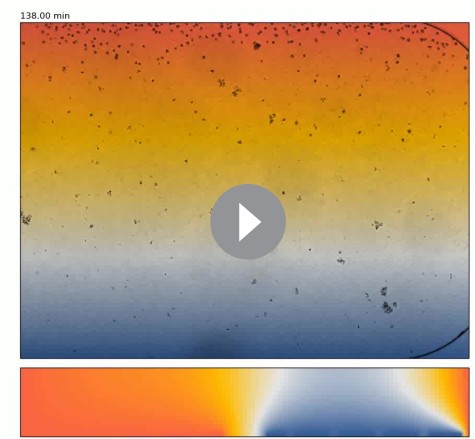

**Video 1.** Experimental videos of aerotaxis (top) and oxygen gas simulation (bottom) (as in *Figure 6*). Experimental videos are colored by the output of the gas simulation. Colonies migrating from one side to other after a swap of nitrogen and air (138–148 min. in experiment of *Figure 3*).

*Figure 5a* shows the angular distribution data during the swaps, where, for the purposes of displaying all the data in a single graph, we have let $\theta \to -\theta$ for times when the oxygen gradient were pointing down. This distribution favors the up-direction $\theta = \pi/2$. More interesting is the distribution of reorientations. For this we define the angle turned by a colony in a time $\Delta t$ as $\Delta\phi = |\theta(t + \Delta t) - \pi/2| - |\theta(t) - \pi/2|$ such that it is positive if the turn is in the direction of the gradient and negative otherwise, and choose $\Delta t$ low enough that $-\pi < \Delta\phi < \pi$. *Figure 5b* shows this distribution. The distribution is centred on zero, revealing that the colonies do as many turns in the wrong direction as in the correct direction. This indicates that the colonies navigate by a *stochastic* strategy.

## Model

We study the spatio-temporal evolution of the choanoflagellate colony population within the observation chamber with the Keller-Segel model (*Keller and Segel, 1971*), which has broad applicability for taxis (*Tindall et al., 2008*). In describing the phenomenon of aerotaxis the two quantities of interest are the colony population density $\rho(\boldsymbol{x}, t)$, and the oxygen concentration $c(\boldsymbol{x}, t)$. The former obeys the Keller-Segel equation

$$\frac{\partial \rho}{\partial t} = D\nabla^2\rho - \nabla \cdot (\mathcal{V}[c]\,\rho) \qquad (1)$$

and the latter follows the diffusion equation,

$$\frac{\partial c}{\partial t} = \nabla \cdot (D_c \nabla c). \qquad (2)$$

The functional $\mathcal{V}$ specifies the population drift velocity's dependency on the local oxygen concentration, *i.e.* the drift of cells due to taxis. For a fixed response, the functional would equal a constant $\mathcal{V} = \mathbf{v}_{\mathrm{drift}}$. $D$ and $D_c(\boldsymbol{x})$ are the colony and oxygen diffusion constants, the latter of which varies with position inside the microfluidic device, with values $D_{c,\mathrm{PDMS}} = 3.55 \times 10^{-3}\,\mathrm{mm}^2/\mathrm{s}$ in PDMS (*Cox and Dunn, 1986*) and $D_{c,\mathrm{water}} = 2.10 \times 10^{-3}\,\mathrm{mm}^2/\mathrm{s}$ in water (*Cussler, 1997*).

Using in-house software, we solve *Equation 2* on a cross-section of the microfluidic device with time-dependent boundary conditions corresponding to the experimental protocols. Gas channels with oxygen flowing have the condition $c = 20\%$ (for theory the unit of oxygen is not important and we find percentage to be the most intuitive measure). For channels with nitrogen flowing $c = 0$. Glass interfaces have no-flux conditions $\hat{\mathbf{n}} \cdot \nabla c = 0$, where $\hat{\mathbf{n}}$ is the surface normal.

A snapshot from the numerical studies is shown in *Figure 6a* at a time following a swap of

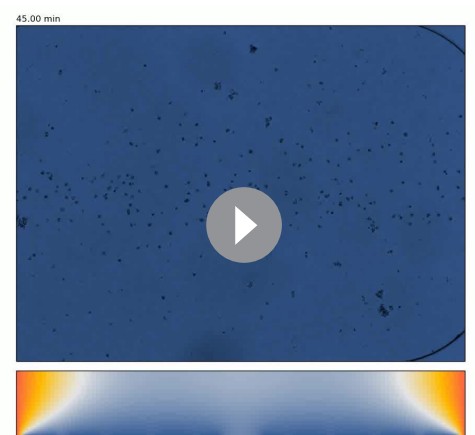

**Video 2.** Colonies migrating from the middle to the sides after a change from nitrogen only in the two gas channels to air only (45–55 min in experiment of *Figure 3*.

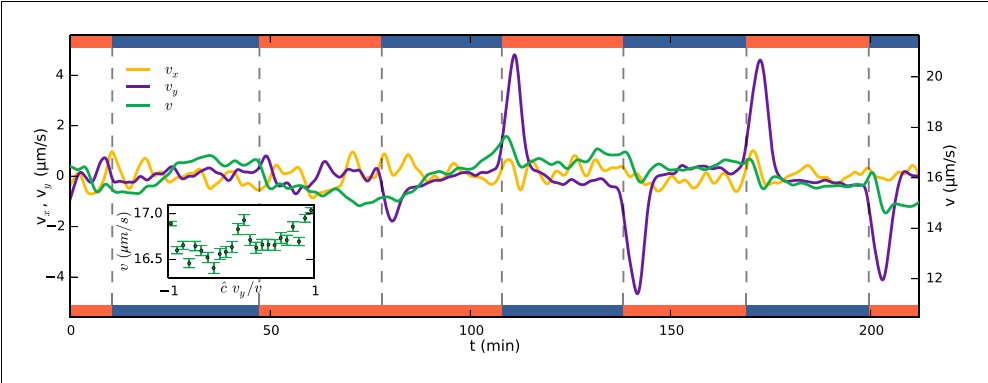

**Figure 4.** Running mean velocity statistics, showing that the primary mechanism of aerotaxis is not by modulation of swimming speed. Evolution of mean speed (green, right axis) and velocity in the $x$-direction (yellow, left axis) and $y$-direction (purple, left axis), $y$ being along the gradient of oxygen. Left and right axes have equal ranges. Side bars indicate gas flowing, oxygen (red) or nitrogen (blue). The peaks of $v_y$ do not quite reach the true drift velocity due to smoothing of the curves. Inset shows the speed as function of alignment with the gas gradient $\hat{c}\, v_y/v$ at times after a swap. $\hat{c} = 1$ if the gradient is up and $= -1$ if down.

The following figure supplement is available for figure 4:

**Figure supplement 1.** Example tracks.

nitrogen and oxygen channels, thus showing residual oxygen above the channels now filled with nitrogen and vice versa. The simulation can now be evaluated at the position of the observation chamber. Note that the no-flux conditions at the glass interface render the concentration gradients in the $z$-direction very small, so the precise height of evaluation is not significant. The simulation with boundary conditions corresponding to those in the experiment of *Figure 3c* is shown in *Figure 6b*. To a very good approximation the concentration field is constant along the $x$-direction. *Figure 6c* shows the oxygen field evaluated at $y = -250$ μm (red curve). Neglecting the consumption of oxygen by the colonies, these results provide the input concentration field $c$ for the Keller-Segel model.

The simplest and most widely-used response functional in the Keller-Segel model is linear in spatial gradients, $\mathcal{V} = \beta\nabla c(\boldsymbol{x}, t)$, where $\beta$ is termed the *taxis coefficient*. Such a response can, however, reach unrealistic drift velocities, *i.e.* higher than the swimming velocity, if the oxygen gradients are

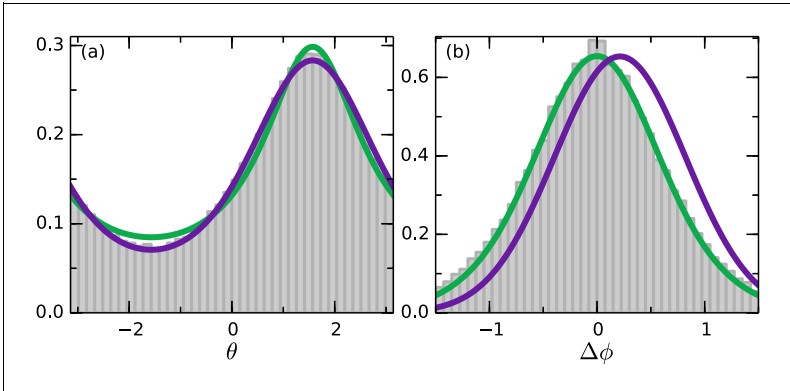

**Figure 5.** Angle statistics. Experimental data in grey bars. Deterministic model in purple and stochastic in green. (a) Distribution of $\theta$. $\theta = \pi/2$ is along the gradient. (b) Change in angle $\Delta\phi$ for $\Delta t = 0.65$ s. Positive change corresponds to a turn towards the gradient. Deterministic parameters: $\epsilon_d = 0.28\,\text{s}^{-1}$, $d_r = 0.52\,\text{s}^{-1}$. Stochastic parameters: $\epsilon_s = 0.55$, $d_r = 0.33\,\text{s}^{-1}$.

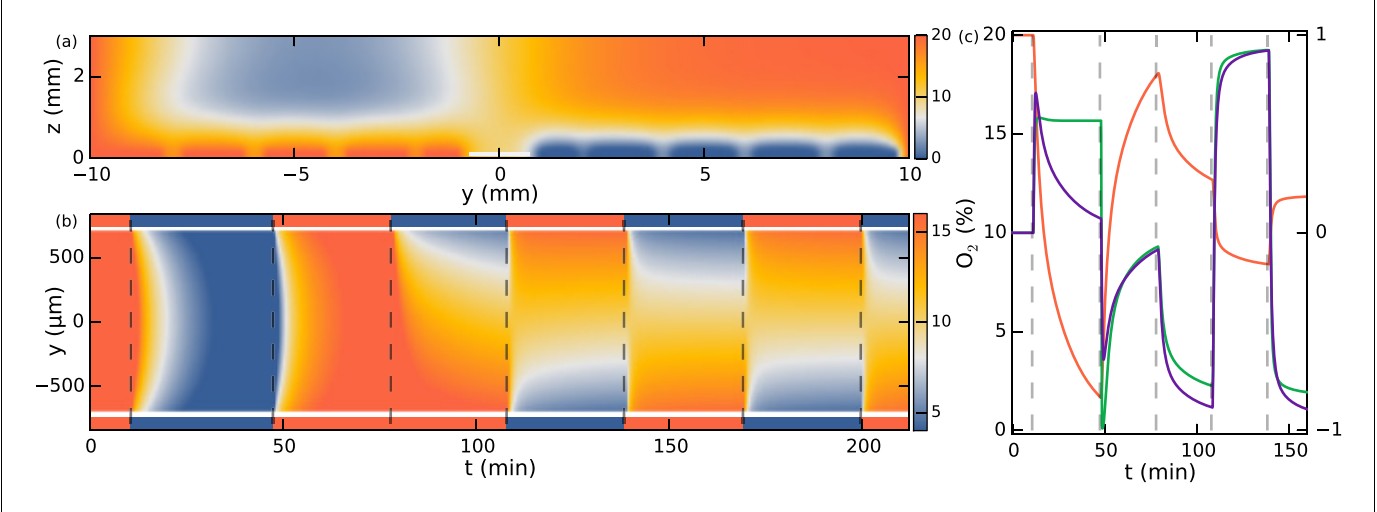

**Figure 6.** Simulation of oxygen concentration in microfluidic device. (a) Simulation of 2D cross-section of the device. Oxygen concentration boundary conditions are imposed at the gas channel positions. Snapshot shows $t = 110$ min, ~1.5 min after the swap. White line indicates evaluation location at the observation chamber. (b) Evolution of oxygen concentration at $z = 100$ μm. (c) Simulation at $y = -250$ μm. Oxygen percentage in red (left axis), and spatial gradient in purple normalized to fit in $[-1, 1]$ (right axis), response function $\tanh(\alpha\nabla c(\mathbf{x}, t)/c(\mathbf{x}, t))$ in green (right axis).

large. This defect can be eliminated by various functional forms (*Tindall et al., 2008*), such as the choice

$$\mathcal{V}[c] = v_{\mathrm{drift}} \tanh(\alpha|\nabla c|)\frac{\nabla c}{|\nabla c|}, \tag{3}$$

where $v_{\mathrm{drift}}$ must be smaller than the swimming velocity. This choice behaves linearly for small gradients, but tends asymptotically to a maximum value for large gradients. Most of the behaviour of *Figure 3c* can be explained by this model (See *Figure 3—figure supplement 1*), but not the section where nitrogen is flowing in both side channels. The reason is the aforementioned asymmetry between the sensing-signalling in low versus high oxygen concentrations, which we argued may be explained by relative gradient sensing, also known as logarithmic sensing since $\nabla \log c = \nabla c/c$.

To examine quantitatively this hypothesis we consider

$$\mathcal{V}[c] = v_{\mathrm{drift}} \tanh\left(\alpha\left|\frac{\nabla c}{c}\right|\right)\frac{\nabla c}{|\nabla c|}. \tag{4}$$

For unidirectional gradients (say, in the $y$-direction) this expression reduces to $\mathcal{V}[c] = v_{\mathrm{drift}} \tanh(\alpha c_y/c)\hat{\mathbf{y}}$, where $c_y = \partial c/\partial y$. Since the logarithmic sensing cannot be maintained at infinitely small concentrations, there must naturally be some lower cutoff to this expression in absolute concentration levels. Nonetheless, from a modelling perspective we can ignore this for the present experiments with nitrogen-only sections lasting less than 1.5 hr as discussed. *Figure 6c* shows that the spatial oxygen gradient $c_y$ (purple curve) compares to $\hat{\mathbf{y}} \cdot \mathcal{V}[c]$ of *Equation 4* (green curve) at all times except in the nitrogen-only section, where the log-response function does not decay towards zero. A positive value of the response function means a positive ($y$) drift velocity.

The Keller-Segel equation with log-sensing is able to explain all sections of the experiment, as demonstrated in *Figure 3d*. The parameters obtained from a numerical fit include the drift velocity $v_{\mathrm{drift}} = 5.2$ μm/s (which should be compared to the ensemble average swimming speed $v = 16.5$ μm/s, the speed averaged over all colonies) and the diffusion constant $D = 865$ μm$^2$/s. Using the ensemble-averaged speed, the diffusion constant can be related to an effective rotational diffusion constant $D_r = v^2/2D = 0.16\,\mathrm{s}^{-1}$.

The parameter $v_{\mathrm{drift}}$ represents the coupling between the oxygen-gradient response and the resulting population drift, but it is a purely phenomenological quantity in which the underlying

microscopic mechanism of aerotaxis is hidden. To explain the origin of $v_{\text{drift}}$ we must consider the navigation strategy.

To distinguish deterministic and stochastic strategies we introduce two effective models and in the following consider them in the context of a constant oxygen gradient along the $y$-axis, but the generalization is immediate. We furthermore ignore translational diffusion due to thermal noise, since this contribution is orders of magnitude smaller than that of active diffusion. Thus in a quasi-2D system, an organism's path is described by

$$\mathrm{d}\boldsymbol{x} = \begin{pmatrix} \cos\theta(t) \\ \sin\theta(t) \end{pmatrix} v\,\mathrm{d}t, \tag{5}$$

where $\theta$ is the instantaneous swimming direction, and we choose motion along the positive $y$-axis ($\theta = \pi/2$) to be toward the attractant.

In the deterministic model, the organisms actively steer towards the gradient. We model this with the Langevin equation

$$\mathrm{d}\theta = \epsilon_d \cos\theta\,\mathrm{d}t + \sqrt{2d_r}\,\mathrm{d}W(t), \tag{6}$$

where $W(t)$ is a Wiener process for which $\langle \mathrm{d}W(t)\,\mathrm{d}W(t')\rangle = \delta(t - t')$. This process is illustrated in *Figure 7a*, yellow arrows showing $\epsilon_d \cos\theta\,\Delta t$ and purple arrows $\sqrt{2d_r}\,\Delta W$. For the stochastic model we take a 'continuous' version of run-and-tumble, in which the rotational diffusion is modulated

$$\mathrm{d}\theta = \sqrt{2d_r(1 - \epsilon_s \sin\theta)}\,\mathrm{d}W(t), \tag{7}$$

where $-1 \leq \epsilon_s \leq 1$ and the multiplicative noise is interpreted in the Itō sense. This process is illustrated in *Figure 7b*. In both models, $\epsilon$ is an effective ensemble average response. To compare to the experiments, $\epsilon$ should be replaced by a function coupled to the gas concentration field through $\mathcal{V}[c]$, thus coupling to *Equation 5*. The steady state gradient in the experiment is approximately linear, and thus as a first approximation *Equations 6, 7* should describe the angle statistics in the time between the swap of gas channels and reaching the opposite side.

In our experiments, after a flip in the oxygen gradient direction, the colonies reach the opposite wall in a time comparable to that for oxygen to diffuse across the chamber. Thus, a true steady state is not reached, but during the intermediate times a steady state approximation is accurate. Solving the Fokker-Planck equations corresponding to the systems *Equations 6 and 7* in steady-state, we

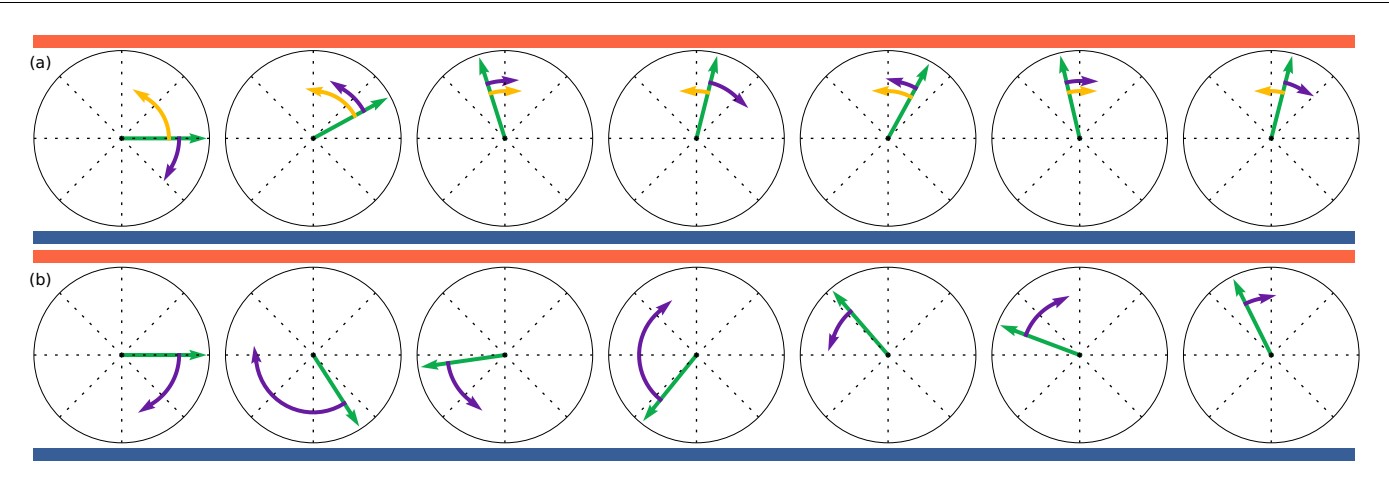

**Figure 7.** Illustration of deterministic and stochastic strategies based on discretised simulations with exaggerated steps. Time evolves from left to right. Orientations, shown by green arrows, are trying to align to up-motion, $\theta = \pi/2$, indicated by red (oxygen) at the top and blue (nitrogen) at the bottom. (a): Deterministic strategy, described by *Equation 6*. Deterministic part in yellow and stochastic part in purple. The deterministic part is always in the correct direction. (b): Stochastic strategies, described by *Equation 7*. All steps are stochastic, but largest when furthest away from $\theta = \pi/2$.

obtain theoretical angle distributions (see Materials and methods). Both models are able to describe the data in *Figure 5a* well, with the deterministic model (purple) fitting the down-gradient swimming best, and the stochastic model (green) fitting the up-gradient swimming best. Both fits involve a single parameter, $k_d = \epsilon_d/\sqrt{2d_r}$ in the deterministic model and $\epsilon_s$ in the stochastic one.

We now move beyond steady-state distributions to examine the detailed statistics of the trajectories themselves. We recall our definition of the angle turned by a colony in a time $\Delta t$ as $\Delta\phi = |\theta(t+\Delta t) - \pi/2| - |\theta(t) - \pi/2|$ such that it is positive if the turn is in the direction of the gradient and negative otherwise. The *Equations (6) and (7)* imply distributions of $\Delta\phi$ (see Materials and methods). *Figure 5b* shows the best fit of both models to the data. For the stochastic model (green) $\epsilon_s$ is known and the fit is in $d_r$ and matches well. The deterministic (purple) is constrained by $k_d = \epsilon_d/\sqrt{2d_r}$, and the fit can be done in $d_r$ as well. We see clearly that the deterministic model does not provide a satisfactory fit to the data. In detail, the value of $\epsilon_d$ needed to fit the data in *Figure 5a* shifts the mean of $p_d(\Delta\phi)$ in the positive direction. This result persists with any amount of smoothing applied to the data, averaging out active rotations. We thus conclude that the colonies navigate by a stochastic strategy, and that the ensemble angle statistics can be captured by this simple model.

The navigation strategy model must be consistent with the Keller-Segel population dynamics model. Having shown that the data favor a stochastic model, we may now couple *Equation 7* to *Equation 5* and let the Fokker-Planck equation (Materials and methods) replace the Keller-Segel model. Such an approach leads to similar results as *Figure 3d*. Furthermore, we now recognize that the Keller-Segel model is a quasi-stationary approximation and we can calculate the stationary-approximation drift velocity

$$v_{\text{drift}} = v\langle\sin\theta\rangle = v\left[1/\epsilon_s - \sqrt{1/\epsilon_s^2 - 1}\right]. \tag{8}$$

In other words $\epsilon_s = (2vv_{\text{drift}})/(v^2 + v_{\text{drift}}^2)$ is the ratio of the squared geometric mean to the quadratic mean of the average and drift velocities. For the fitted $\epsilon_s = 0.55$, $v_{\text{drift}} \approx 0.3\,v \approx 5$ μm/s, consistent with the fitted value in the Keller-Segel model.

## Discussion

We have shown that colonies of *S. rosetta* can navigate along gradients of oxygen, thus exhibiting positive aerotaxis. The cells navigate along relative oxygen gradients and the navigation strategy is stochastic in nature, achieved by modulating not speed but direction of swimming.

The experimental observation that choanoflagellates are aerotactic raises a number of questions. One concerns the actual sensing mechanism, and how this compares to those of animals, given the close evolutionary relationship between choanoflagellates and metazoans. In animals, oxygen concentrations can be sensed by the highly-conserved hypoxia-inducible factor (HIF) transcription factor pathway (*Loenarz et al., 2011*; *Kaelin and Ratcliffe, 2008*; *Rytkönen et al., 2011*). At normal oxygen levels, the activity of specific prolyl-hydroxylase (PHD) enzymes labels the HIF protein complexes for degradation. At low oxygen levels, however, PHD activity is inhibited, leading to elevated HIF levels. The transcription factor activity of HIF up-regulates expression of genes involved in hypoxia response (e.g. glycolysis enzymes) for survival in low-oxygen conditions (*Greer et al., 2012*). The genes involved in HIF signalling are widespread in metazoans, with evidence suggesting that some components of this pathway are descended from prokaryotic ancestors (*Scotti et al., 2014*).

Preliminary experiments involving exposing *S. rosetta* cultures to DMOG (an inhibitor of the prolyl-hydroxylase step in the HIF pathway [*Fong and Takeda, 2008*]) had no observable effect on aerotaxis within our experimental system (data not shown). This is perhaps unsurprising, given that choanoflagellates lack some important components of the HIF pathway (*Rytkönen et al., 2011*). Additionally, the aerotactic response observed here is acute, within a timeframe on the order of seconds to minutes, rather than the being a longer-term response to lowered oxygen levels by modulation of gene expression. In certain aerotactic bacteria, it is known that oxygen concentrations are measured indirectly via energy-sensing, which can then influence the rotation of the bacterial flagella between running and tumbling (*Taylor et al., 1999*). An analogous mechanism may be at play in choanoflagellates, possibly via reactive oxygen species (*Cash et al., 2007*) and oxygen-sensitive ion channels (*Lahiri et al., 2006*; *Ward, 2008*) modulating the beating rate of the flagella of each *S.*

*rosetta* cell. Thus the question of the molecular and cellular mechanisms underpinning aerotaxis in choanoflagellates is a promising avenue for further research.

Oxygen is implicated as having an important influence on animal evolution (*Nursall, 1959*; *Lenton et al., 2014*), with some hypotheses that the emergence of complex animal ecosystems (*Sperling et al., 2013*) were only triggered when oxygen levels rose above certain thresholds. Sponges, believed to be the most basal animal group, were recently shown to have very low oxygen requirements, disputing the importance of oxygen in early animal evolution (*Mills et al., 2014*). Our results raise the possibility that the common ancestor from which choanoflagellates and animals evolved was aerotactic, and that oxygen sensing and responding have thus been under strong selection throughout the holozoans. If the ancestors of animals and choanoflagellates were strongly aerotactic, this would be indicative of the importance of oxygen as a resource during the Precambrian (*Lyons et al., 2014*). Equally, it could be the case that choanoflagellates themselves have evolved aerotaxis after the split from the animal stem lineage. To answer this question, the oxygen requirements and aerotactic capabilities of other opisthokont groups, e.g. ichthyosporeans, filasterians (*Ruiz-Trillo et al., 2008*; *Sebé-Pedrós et al., 2013*), as well as basal animals such as sponges, ctenophores and placozoans (*Dunn et al., 2008*; *Ryan et al., 2013*; *Pisani et al., 2015*; *Whelan et al., 2015*), requires investigation. Such an analysis can be further complimented by determining the oxygen sensing and signalling mechanisms across the opisthokont phylogeny.

If aerotaxis was key to the ancestral unicellular holozoan, it is crucial that the evolution to multicellularity did not hinder this ability. Our experimental results show that both the unicellular and colonial morphotypes of *S. rosetta* can perform efficient aerotaxis and navigation in general, despite lacking coordination between the constituent cells of the colony (*Kirkegaard et al., 2016*). Therefore, an evolutionary transition to a multicellularity resembling the unicellular-to-colonial transition in *S. rosetta* (*Dayel et al., 2011*) would not require additional cell-cell communication mechanisms to coordinate navigation. Not only would this allow aerotaxis towards oxygen, but also taxis in response to other stimulants, such as bacterial signals (*Woznica et al., 2016*). This is a particular feature of the stochastic navigation strategy, which works equally well for both single cells and colonies formed from multiple units of the same cells.

We have described the stochastic strategy of *S. rosetta* in terms of an effective model. The effective bias parameter $\epsilon_s$ is the result of flagella modulation. Flagella can be imaged on colonies stuck to the microscope slide (*Kirkegaard et al., 2016*), but measuring directly the flagella modulation is challenging, since the oxygen changes that can be induced in a microfluidic device are on a time scale of minutes, whereas the flagella beating is on a time scale of tens of milliseconds (*Kirkegaard et al., 2016*). The swimming trajectories suggest that this modulation is a mixture of many types, ranging from slow to vigorous, as also seen in other organisms (*Jikeli et al., 2015*), but direct imaging is needed to make more quantitative statements.

We have shown that in spatio-temporal varying environments, considerations of population dynamics can distinguish between linear- and logarithmic-sensing mechanisms, and concluded that choanoflagellates do logarithmic sensing. The fact that conclusions of logarithmic sensing can be made from analysis of the population dynamics alone shows that individual tracks are not needed and thus allows for such analysis in dense experiments. Logarithmic sensing is a key attribute for survival: it allows sensing of and reacting to gradients in very low concentration environments, while still being able to effectively navigate along large gradients. Logarithmic sensing has also been experimentally observed for other species, *e.g.* bacteria (*Mesibov et al., 1973*; *Kalinin et al., 2009*).

With logarithmic sensing, cells only navigate along oxygen gradients that are significant compared to the absolute concentration. This is a well-known phenomenon also from human behaviour, where, for instance, dim lights are seen only when it is dark and weak sounds heard only when it is quiet. It is known as Weber's law (*Ross, 1996*), which states that the magnitude of just-noticeable differences of a stimuli is proportional to the stimuli magnitude itself. This is closely related to the Weber-Fechner law, stating that stimuli magnitude grows logarithmically with the actual signal, which we have found to be in agreement with the experimental observation of *S. rosetta* aerotaxis.

For microorganisms, a Weber lower limit can be understood, at least partly, as a physical limitation. Because of thermal fluctuations, the error on any concentration gradient measurement increases with the local absolute concentration (*Berg and Purcell, 1977*; *Endres and Wingreen, 2008*), and thus it immediately follows that the limit of just-noticeable gradients must decrease with absolute concentration. Quantitatively, the noise scales as $\sim c^{1/2}$ in the absolute concentration

(*Endres and Wingreen, 2008*). Such a scaling is not observed for bacteria's just-noticeable limits, where instead Weber's law hold (*Mesibov et al., 1973*). Sensor adaptation enables the bacteria to have the linear scaling $\sim c$ (*Sourjik and Wingreen, 2012*) leading to logarithmic sensing and high dynamic range, but it is nonetheless the case that for purely physical reasons a lower limit scaling with concentration must be present. The fitting of the Keller-Segel model to the population data were optimal precisely for relative gradient sensing. Sensing functionals such as $\nabla c/\sqrt{c}$ or $\nabla c/c^2$ only decreased the fit quality. This implies that Weber's law and logarithmic sensing, at least to a good approximation, is occurring in chaonoflagellates. This scaling could be further studied by measuring the aerotactic response to an order-of-magnitude range of concentration gradients under a range of absolute concentrations. Precision control of gas mixtures would allow the extraction of any biological deviations from Weber's law and potentially reveal a transition to the physical limit of $\sqrt{c}$ scaling for very low concentrations, *i.e.* at the limit of sensing.

## Materials and methods

### Culturing *S. rosetta*
*S. rosetta* were cultured polyxenicly in artificial seawater (36.5 g/L Marin Salts [Tropic Marin, Germany]) with organic enrichment (4 g/L Proteose Peptone [Sigma-Aldrich, USA], 0.8 g/L Yeast Extract [Fluka Biochemika]) at 15 µl/mL, and grown at 23°C, split weekly. Cultures were centrifuged to reach the high concentrations, $\sim 5 \times 10^6 \, \mathrm{ml}^{-1}$, used in experiments.

### Microfluidic device & imaging
Microfluidic devices were manufactured using standard soft-lithography techniques. The master was produced by spinning SU8-2075 (MicroChem, USA) at 1200 rpm to a thickness of ~115 µm. Chambers were cast in PDMS, Sylgard 184 (Dow Corning, USA), and plasma etched to the glass slides. Cultures were concentrated by centrifugation before loaded into the device. Gas cylinders containing pure nitrogen and air (20% oxygen) were connected via a system of valves to the gas channels of the device. Experiments were filmed (Imaging Source, Germany) in bright field at 10 fps on an inverted Zeiss LSM 700 Microscope.

### Data processing
To track colonies, we first generated a running-median video, where each pixel in each frame of the experimental video is the median of that pixel taken over the neighbouring ~2.5 s of video. This method extracts an estimated background, *i.e.* a video without the colonies present. Subtracting this from the original video, the resulting video contains only the colonies and noise. Band-pass filters were used to remove the noise, and finally the colony positions were found by locating local maxima in the Gaussian filtered video.

Density distributions were estimated by first calculating

$$\eta(y) = \frac{\sum_i \exp(-(y_i - y)^2/2\sigma^2)}{\int_{y_0}^{y_1} \exp(-(y' - y)^2/2\sigma^2)\, \mathrm{d}y'}, \tag{9}$$

where $y_i$ are the tracked positions, $\sigma$ a standard deviation of separation, and the denominator adjusts for boundary effects. Hereafter $\rho(y) = \eta(y)/\int_{y_0}^{y_1} \eta(y')\, \mathrm{d}y'$ is the normalized density.

For velocity and angle statistics, the tracked positions were linked solely by proximity. If the image analysis algorithm failed to identify a given colony over fewer than three successive frames the integrity of the track was preserved by keeping a running memory. After the trajectories were obtained, spurious trajectories less than three frames in length were removed. Examples of trajectories are shown in *Figure 4—figure supplement 1*. The final tracks contain ~150 trajectories in each frame, varying slightly over the course of the experiment due to loss of colonies in the tracking and swimming in and out of observation chamber.

Oxygen diffusion and Keller-Segel equations were numerically solved using in-house software with finite difference spatial discretisation and implicit time-stepping.

## Effective stochastic models

Given the stochastic dynamics (6) for individual particles following a deterministic strategy, the probability distribution function $p_d(\theta, t)$ for the population obeys the Fokker-Planck equation

$$\frac{\partial p_d}{\partial t} = d_r \frac{\partial^2 p_d}{\partial \theta^2} - \epsilon_d \frac{\partial}{\partial \theta}(\cos(\theta) p_d). \tag{10}$$

The steady state distribution is found to be a von-Mises distribution

$$p_d(\theta) = \frac{1}{2\pi I_0(\epsilon_d/\sqrt{2d_r})} \exp\left(\frac{\epsilon_d \sin\theta}{\sqrt{2d_r}}\right), \tag{11}$$

where $I_0$ is the modified Bessel function of order zero. On the other hand, for the Fokker-Planck equation for the stochastic model,

$$\frac{\partial p_s}{\partial t} = \frac{\partial^2}{\partial \theta^2}(d_r(1 - \epsilon_s \sin\theta) p_s), \tag{12}$$

we find the steady-state distribution

$$p_s(\theta) = \frac{1}{2\pi} \frac{\sqrt{1 - \epsilon_s^2}}{1 - \epsilon_s \sin\theta}. \tag{13}$$

In the time right before a channel swap, we have $p(\theta) \approx 1/2\pi$, since the colonies stay near the wall. In the deterministic model, for small $\Delta t$, $\Delta\phi = |\theta(t + \Delta t) - \pi/2| - |\theta(t) - \pi/2|$ is composed of deterministic $\delta = \epsilon_d \cos(\theta)\Delta t$ and stochastic $\xi = \sqrt{2d_r}\Delta W$, where $\Delta W \sim \sqrt{\Delta t}$. Since we are assuming $p(\theta) = 1/2\pi$, we have $p(\delta) = 1/\pi\sqrt{(\epsilon_d\Delta t)^2 - \delta^2}$ for $\delta \in (-\epsilon_d\Delta t; \epsilon_d\Delta t)$. The distribution of $\Delta\phi = |\delta| + \xi$ is then found as the convolution

$$p_d(\Delta\phi) = \int_0^{\epsilon_d\Delta t} \frac{\exp(-(\Delta\phi - \delta)^2/4d_r\Delta t)}{\sqrt{\pi^3 d_r \Delta t \left[(\epsilon_d\Delta t)^2 - \delta^2\right]}} \, d\delta, \tag{14}$$

which can be evaluated numerically by Gaussian quadrature. In the stochastic model there is only $\xi = \sqrt{2d_r(1 - \epsilon_s \sin\theta)}\Delta W$, but this is conditional on $\theta$. We marginalize for the final distribution

$$p_s(\Delta\phi) = \int_{-\pi}^{\pi} \frac{\exp(-\Delta\phi^2/4d_r(1 - \epsilon_s \sin\theta)\Delta t)}{\sqrt{16\pi^3 d_r(1 - \epsilon_s \sin\theta)\Delta t}} \, d\theta. \tag{15}$$

## Acknowledgements

Work supported by the EPSRC and St John's College (JBK), ERC Advanced Investigator Grant 247333 and a Wellcome Trust Senior Investigator Award.

## Additional information

### Funding

| Funder | Grant reference number | Author |
|---|---|---|
| European Research Council | 247333 | Ambre Bouillant<br>Alan O Marron<br>Raymond E Goldstein |
| Wellcome Trust | 097855 | Alan O Marron<br>Kyriacos C Leptos<br>Raymond E Goldstein |
| Engineering and Physical Sciences Research Council | EP/M017982/1 | Julius B Kirkegaard<br>Raymond E Goldstein |

The funders had no role in study design, data collection and interpretation, or the decision to submit the work for publication.

## Author contributions
JBK, Conception and design, Acquisition of data, Analysis and interpretation of data, Drafting or revising the article; AB, Conception and design, Acquisition of data, Analysis and interpretation of data; AOM, KCL, REG, Conception and design, Analysis and interpretation of data, Drafting or revising the article

## Author ORCIDs
Raymond E Goldstein, http://orcid.org/0000-0003-2645-0598

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
