## [Decision Letter]

Thank you for submitting your article "Aerotaxis in the Closest Relatives of Animals" for consideration by *eLife*. Your article has been favorably evaluated by Naama Barkai as the Senior Editor and three reviewers, one of whom, Richard M Berry (Reviewer #1), is a member of our Board of Reviewing Editors.

The reviewers have discussed the reviews with one another and the Reviewing Editor has drafted this decision to help you prepare a revised submission.

Summary:

This manuscript presents experimental evidence and theoretical modelling of the following:

1) Choanoflagellates, small colonial eukaryotes, show taxis towards oxygen.

2) Oxygen responses are proportional to relative rather than absolute oxygen gradients (logarithmic).

3) Taxis is achieved by modulation of swimming direction, not swimming speed.

4) Modulation of direction is stochastic, not deterministic.

The data are convincing, the modelling thorough and appropriate, and the conclusions sound.

Essential revisions:

1) The paper reads as written from a theoretical perspective, with the experimental data as an accessory. This needs to be reversed, the significance of the paper is in the experimental data. Results 1, 3 and 4 above are directly demonstrated by the data of Figure 3, Figure 6 and 8B respectively, and this should not be hidden behind the modelling as it currently is. The asymmetry between all-air and all-nitrogen flows is also evident in Figure 3, and the inference of logarithmic sensing from this is simple to make without quantitative modelling. The data should be described and the inferences drawn, and the modelling relegated to its proper place as quantitative verification of the inferences that can be made directly from the data.

Correspondingly, the data and their analysis need to be described much more thoroughly. The algorithms for obtaining tracks need to be better described than the present "Colonies were tracked using custom in-house software". Also, show some examples of tracks, and give statistics i.e. numbers of tracks used for each result presented, and any selection or other processing of tracks that was done. For example the Discussion states, "The fact that such conclusions can be made from population dynamics alone allows similar analysis to be done in dense populations, where long individual tracking is not possible." – but nowhere in the manuscript is it possible to know whether current experiments were done under this condition or not!

2) The statement and discussion of the significance of the results needs to be altered and expanded.

At present, the entire paper is based on the premise that choanoflagellates are important to study, because, as the closest living unicellular relative of animals, they can provide unique insight into the origin of animal multicellularity. While this might be true, it is a tricky argument to make correctly, and many authors fall into the trap of assuming that choanoflagellates are representative of the actual ancestor of animals. To be clear, choanoflagellates are not representative of the actual ancestor of animals – they diverged from the actual ancestor of animals ~800 million years ago and have been changing (evolving!) all that time. Indeed, these authors make this mistake as well. In last sentence of the Discussion, they say "Our results add to this discussion, in showing that their evolutionary precursors, the choanoflagellates, have evolved to navigate actively towards oxygen." The problem with this logic is that, of course, choanoflagellates are not the evolutionary precursor to animals- those are long extinct. Without looking at aerotaxis in a phylogenetic context, including many unicellular linages of opisthokonts, we don't know if this particular lineage of choanoflagellates evolved aerotaxis prior to the split with animals, when oxygen levels were a fraction of what they are today, or if they evolved it during the intervening ~800 million years. In fact, a trait like oxygen is a particularly poor candidate for the choanoflagellate-as-protoanimal comparison (in contrast to, say, a mechanism of cellular signaling that is highly conserved), since oxygen levels have changed dramatically during the period of time after which choanoflagellates split from the animal stem lineage, and the ability to respond to oxygen is almost certainly a trait that is under strong selection. Thus, while I completely believe the authors that modern-day *S. rosetta* is positively aerotactic, this work does not tell us anything about whether the ancestor of animals was aerotactic.

Instead, the multicellularity argument could be moderated and still be interesting. For example, while making it clear that this work does not tell us whether aerotaxis evolved prior to the origin of animals, it shows that it is a possibility. The authors should suggest how, with further experiments in phylogenetically diverse unicellular lineages diverging from the animal stem group, researchers could test this hypothesis using pretty standard phylogenetic tools.

They should also spend some energy in the Discussion explaining why it matters if the unicellular ancestor of animals is aerotactic. We already know that they were capable of aerobic metabolism – how would it change our view of animal multicellularity to know that they swim towards this key resource? Since they lose the intrinsic importance of a discovery (e.g., that the ancestor of animals was aerotactic), it is important to explain why the general idea is important.

One possibility might be that the importance lies in showing that a form of apparently disorganized multicellularity, which may or may not model early metazoan life, but models it as well as anything that we are aware of, is capable of any taxis. Although aerotaxis is relatively common, it is still surprising that *S. rosetta* can do it, because it does not coordinate its flagella (cf. the Volvocales, in which there is a lot of high profile work on e.g. phototaxis). This suggests possibilities for how early multicellularity may have worked.

3) Logarithmic sensing requires some simple explanation for non-mathematical readers. Simply put, response is proportional to relative rather than absolute concentration gradient. This result requires more discussion. Give more examples of logarithmic sensing from literature. Are there any exceptions, where sensing is linear, or is logarithmic sensing universal?

4) The entire evidence for the logarithmic response is the persistence of a band of choanoflagellates in the center of the device as oxygen is slowly depleted. This one observation is interpreted a quite complicated model, which has some parametric freedom (although the model fitting is well justified and reasonable). Can one design a different control experiment in which logarithmically sensing cells are predicted to behave differently than cells that have the flux-response given in Equation (3)? E.g. some kind of pulse chase in which they follow a declining flux gradient. It is difficult to imagine how this experiment can be performed, but the point is that the fact that one observation is hard to predict with an existing model only suggests possibilities for alternate models. Additional experiments allow these possibilities to be explored.

---

## [Author Response]

*[…] Essential revisions:*

*1) The paper reads as written from a theoretical perspective, with the experimental data as an accessory. This needs to be reversed, the significance of the paper is in the experimental data. Results 1, 3 and 4 above are directly demonstrated by the data of Figure 3, Figure 6 and 8B respectively, and this should not be hidden behind the modelling as it currently is. The asymmetry between all-air and all-nitrogen flows is also evident in Figure 3, and the inference of logarithmic sensing from this is simple to make without quantitative modelling. The data should be described and the inferences drawn, and the modelling relegated to its proper place as quantitative verification of the inferences that can be made directly from the data.*

Thank you for this comment. We have now reversed the structure of the paper and discussed all experimental results before ending with a single section of modelling. As we wish to reach as broad an audience as possible, this change will prevent readers not familiar with the theory from having to look for the experimental results between the model details. We believe this rearrangement has indeed improved the manuscript's readability.

*Correspondingly, the data and their analysis need to be described much more thoroughly. The algorithms for obtaining tracks need to be better described than the present "Colonies were tracked using custom in-house software". Also, show some examples of tracks, and give statistics i.e. numbers of tracks used for each result presented, and any selection or other processing of tracks that was done. For example the Discussion states, "The fact that such conclusions can be made from population dynamics alone allows similar analysis to be done in dense populations, where long individual tracking is not possible." – but nowhere in the manuscript is it possible to know whether current experiments were done under this condition or not!*

The Methods section has now been expanded to include details of the tracking algorithm and track statistics. We have also added a supplementary figure showing examples of tracks. The sentence quoted above was meant to underlie that our conclusion of logarithmic sensing was not based on individual tracks. We merely wanted to point out that no matter how concentrated our suspensions had been, we could have drawn our conclusion based on the population dynamics alone. We understand, nevertheless, how in its current form the sentence implies that the suspension was not concentrated and we have thus rephrased the sentence. In any case, the tracking details now added to the Methods section should now fully specify our experiments. We thank the reviewers for pointing this out.

*2) The statement and discussion of the significance of the results needs to be altered and expanded.*

*At present, the entire paper is based on the premise that choanoflagellates are important to study, because, as the closest living unicellular relative of animals, they can provide unique insight into the origin of animal multicellularity. While this might be true, it is a tricky argument to make correctly, and many authors fall into the trap of assuming that choanoflagellates are representative of the actual ancestor of animals. To be clear, choanoflagellates are not representative of the actual ancestor of animals – they diverged from the actual ancestor of animals ~800 million years ago and have been changing (evolving!) all that time. Indeed, these authors make this mistake as well. In last sentence of the Discussion, they say "Our results add to this discussion, in showing that their evolutionary precursors, the choanoflagellates, have evolved to navigate actively towards oxygen." The problem with this logic is that, of course, choanoflagellates are not the evolutionary precursor to animals- those are long extinct. Without looking at aerotaxis in a phylogenetic context, including many unicellular linages of opisthokonts, we don't know if this particular lineage of choanoflagellates evolved aerotaxis prior to the split with animals, when oxygen levels were a fraction of what they are today, or if they evolved it during the intervening ~800 million years. In fact, a trait like oxygen is a particularly poor candidate for the choanoflagellate-as-protoanimal comparison (in contrast to, say, a mechanism of cellular signaling that is highly conserved), since oxygen levels have changed dramatically during the period of time after which choanoflagellates split from the animal stem lineage, and the ability to respond to oxygen is almost certainly a trait that is under strong selection. Thus, while I completely believe the authors that modern-day S. rosetta is positively aerotactic, this work does not tell us anything about whether the ancestor of animals was aerotactic.*

*Instead, the multicellularity argument could be moderated and still be interesting. For example, while making it clear that this work does not tell us whether aerotaxis evolved prior to the origin of animals, it shows that it is a possibility. The authors should suggest how, with further experiments in phylogenetically diverse unicellular lineages diverging from the animal stem group, researchers could test this hypothesis using pretty standard phylogenetic tools.*

This is a very welcome comment. Indeed, we had not spent much space in the manuscript discussing the evolutionary context of our results, and undoubtedly our last paragraph in the Discussion was inadequate. We are grateful for having this pointed out. Our Discussion has now been significantly expanded and now includes a section on the relation between animals, choanoflagellates and their common ancestor. We furthermore made it clear, as pointed out in the above comment, that phylogenetic experiments and analysis are needed to make conclusions on the aerotactic capabilities of the common ancestor.

*They should also spend some energy in the Discussion explaining why it matters if the unicellular ancestor of animals is aerotactic. We already know that they were capable of aerobic metabolism – how would it change our view of animal multicellularity to know that they swim towards this key resource? Since they lose the intrinsic importance of a discovery (e.g., that the ancestor of animals was aerotactic), it is important to explain why the general idea is important.*

*One possibility might be that the importance lies in showing that a form of apparently disorganized multicellularity, which may or may not model early metazoan life, but models it as well as anything that we are aware of, is capable of any taxis. Although aerotaxis is relatively common, it is still surprising that S. rosetta can do it, because it does not coordinate its flagella (cf. the Volvocales, in which there is a lot of high profile work on e.g. phototaxis). This suggests possibilities for how early multicellularity may have worked.*

In the Introduction we had mentioned this distinction between the Volvocales and *S. rosetta. W*e are grateful to the reviewers for suggesting that we should also include this in discussing why aerotaxis and our results are important. We have now added a section in the Discussion on this.

*3) Logarithmic sensing requires some simple explanation for non-mathematical readers. Simply put, response is proportional to relative rather than absolute concentration gradient. This result requires more discussion. Give more examples of logarithmic sensing from literature. Are there any exceptions, where sensing is linear, or is logarithmic sensing universal?*

With the reorganisation of the paper we now also explain logarithmic sensing in the Results section, before the modelling. We have furthermore included details on the generality of logarithmic sensing and mentioned the Weber-Fechner law in the Discussion, where we also discuss the necessity of some non- linear scaling on purely physical grounds.

*4) The entire evidence for the logarithmic response is the persistence of a band of choanoflagellates in the center of the device as oxygen is slowly depleted. This one observation is interpreted a quite complicated model, which has some parametric freedom (although the model fitting is well justified and reasonable). Can one design a different control experiment in which logarithmically sensing cells are predicted to behave differently than cells that have the flux-response given in Equation (3)? E.g. some kind of pulse chase in which they follow a declining flux gradient. It is difficult to imagine how this experiment can be performed, but the point is that the fact that one observation is hard to predict with an existing model only suggests possibilities for alternate models. Additional experiments allow these possibilities to be explored.*

While our results definitely exclude Eq. (3) as a possible taxis functional, the parametric fit of the Keller- Segel model does not guarantee that Eq. (4) is precisely the correct form. Nonetheless, we have now mentioned in the Discussion that other slight variations give only worse fits, suggesting that indeed Eq. (4) is the correct one to good approximation. To extract the exact biological functional and observe the differences to Eq. (3), variations on our experiment can be performed with different gas mixtures e.g.by performing an experiment with a gradient built up using 0 and 1% oxygen gas cylinders and comparing to an experiment with 19 and 20% oxygen. Under a linear functional these experiments would give the same results, but not under logarithmic sensing. We have now suggested such future experiments in the Discussion.